# Whole-Exome Sequencing Identified Two Novel Pathogenic Mutations in the *PTCH1* Gene in BCNS

**Margit Pál** [1,2,†], **Éva Vetró** [3,†], **Nikoletta Nagy** [1,2,*], **Dóra Nagy** [1,4], **Emese Horváth** [1], **Barbara Anna Bokor** [1], **Anita Varga** [5], **László Seres** [3], **Judit Oláh** [5,6], **József Piffkó** [3] **and Márta Széll** [1,2]

1. Department of Medical Genetics, University of Szeged, 6720 Szeged, Hungary
2. ELKH-SZTE Functional Clinical Genetics Research Group, Eötvös Loránd Research Network, 6720 Szeged, Hungary
3. Department of Oral and Maxillofacial Surgery, University of Szeged, 6725 Szeged, Hungary
4. Institute of Medical Genetics, Kepler University Hospital Med Campus IV, Johannes Kepler University Linz, 4020 Linz, Austria
5. Department of Dermatology and Allergology, University of Szeged, 6720 Szeged, Hungary
6. Department of Oncotherapy, University of Szeged, 6720 Szeged, Hungary
* Correspondence: nagy.nikoletta@med.u-szeged.hu
† These authors contributed equally to this work.

**Abstract:** Basal cell nevus syndrome (BCNS, OMIM 109400) is a familial cancer syndrome characterized by the development of numerous basal cell cancers and various other developmental abnormalities, including epidermal cysts of the skin, calcified dural folds, keratocysts of the jaw, palmar and plantar pits, ovarian fibromas, medulloblastomas, lymphomesenteric cysts, and fetal rhabdomyomas. BCNS shows autosomal dominant inheritance and is caused by mutations in the patched 1 (*PTCH1*) gene and the suppressor of the fused homolog (*SUFU*) gene. In a few cases, variants of patched 2 (*PTCH2*) have been found in patients who met the criteria for BCNS. In an investigation of 11 Hungarian families who fulfilled the diagnostic criteria for BCNS, whole-exome sequencing (WES) and multiplex ligation-dependent probe amplification (MLPA) identified two novel pathogenic variants (c.2994C>A; p.Cys998Ter and c.814_818del; p.Asn272SerfsTer11), one recently identified variant (c.1737_1745del p.Val580_Val582del), and three recurrent disease-causing variants of the *PTCH1* gene with a diagnosis rate of 63.6%. Disease-causing variants were not found for the *SUFU* and *PTCH2* genes. These applied methods could not fully elucidate the genetic background of all the BCNS cases that we investigated. To uncover the missing heritability of BCNS, whole-genome sequencing or an epigenetic approach might be considered in the future.

**Keywords:** basal cell nevus syndrome; Gorlin syndrome; whole-exome sequencing; multiplex ligation-dependent probe amplification; missing heritability

## 1. Introduction

Basal cell nevus syndrome (BCNS, OMIM 109400), also referred to as nevoid basal cell carcinoma syndrome, basal cell carcinoma nevus syndrome, Gorlin syndrome, or Gorlin–Goltz syndrome, is a familial cancer syndrome, which predisposes to overgrowth and tumor formation [1]. BCNS is characterized by skin tumors, along with skeletal, ophthalmologic, and neurologic abnormalities. Specifically, BCNS is characterized by the development of basal cell carcinomas and epidermal cysts of the skin, calcified dural folds, keratocysts of the jaw, palmar and plantar pits, ovarian fibromas, medulloblastomas, lymphomesenteric cysts, fetal rhabdomyomas, and various other developmental abnormalities [1,2]. The high phenotypic variability of BCNS can lead to delayed diagnoses and, subsequently, delayed treatments that greatly increase associated morbidity and even mortality levels [1,2]. BCNS was first described in the 1960s by Drs. Gorlin and Goltz [1]. Over the past 60 years, useful diagnostic criteria have been developed. Currently, a diagnosis of BCNS requires the

presence of two major diagnostic criteria or one major and two minor diagnostic criteria. Major criteria include multiple BCCs or one BCC by an age of 20 years, odontogenic keratocysts of the jaw (as proven by histology), palmar or plantar pitting, bilamellar calcification of the falx cerebri, bifid or fused or splayed ribs, and first-degree relatives with BCNS. Minor criteria include medulloblastoma, increased circumference of the head, congenital malformations (such as frontal bossing, coarse facies, cleft lip and/or palate, and moderate or severe hypertelorism), other skeletal abnormalities (such as Sprengel deformity, marked pectus deformity, and the marked syndactyly of digits), radiologic abnormalities (e.g., bridging of the sella turcica, hemivertebrae, fusion or elongation of vertebral bodies, modeling defects of the hands and feet, or flame-shaped lucencies of the hands or feet), and ovarian and cardiac fibromas [3].

Although the most reported prevalence rates for BCNS are 1 in 40,000–60,000, the true prevalence might be higher, as patients with a mild phenotype might not be diagnosed correctly with BCNS, and, therefore, might not have been included in the statistics. The disease affects both men and women in a nearly equal distribution. All races around the world have been reported to be affected by BCNS, although African Americans and Asians represent only 5% of known cases [1–3].

BCNS exhibits an autosomal dominant mode of inheritance with variable expression and nearly complete penetrance, and molecular alterations in components of the highly conserved sonic hedgehog (SHH) signaling pathway have been implicated in its pathogenesis. Mutations of the patched 1 (*PTCH1*) and the suppressor of fused homolog (*SUFU*) genes have been reported to play an important role in the development of BCNS [4]. Variants of the patched 2 (*PTCH2*) gene have been found in a few patients that met the criteria for BCNS [5,6]. Hot spots for genes and genotype–phenotype relationships have not yet been identified. The proteins coded by these three genes are important members of the SHH signaling pathway. *PTCH1* and *PTCH2* encode transmembrane receptor proteins that recognize signaling proteins of the SHH family, and SUFU is a negative regulator of the SHH signaling pathway. The improper function of the SHH pathway leads to developmental disorders and malignancies, affecting the differentiation processes of embryogenesis, tissue regeneration, and cell division processes [7]. SHH plays an important role in nervous system cell-type specification and in limb patterning [7].

Seventy to eighty percent of patients with BCNS have a familial aggregation of symptoms, and de novo pathogenic variants are responsible for 20–30% of cases. In rare cases, gonadal or somatic mosaicism may occur [8]. In approximately 50–85% of BCNS cases sequence variations of the *PTCH1* gene, in 6–21% duplication or deletion of exon(s) of the *PTCH1* gene, in approximately 5% sequence variation of the *SUFU* gene, and in 1% deletion or duplication of exon(s) of the *SUFU* gene are at the background of the disease [4]. In 15–27% of patients the genetic mechanism causing BCNS remains unknown. In general, BCNS patients with *SUFU* variants have an increased risk of developing medulloblastoma compared to those carrying *PTCH1* mutations [3].

Both PTCH1 and SUFU are also involved in other genetic diseases. Mutations in the *PTCH1* gene have been associated with holoprosencephaly type 7 (OMIM 610828). *SUFU* mutations have been described in familial meningioma (OMIM 607174), desmoplastic medulloblastoma (OMIM 155255), and Joubert syndrome-32 (OMIM 617757), the latter of which is a developmental disorder that is characterized by delayed psychomotor development, intellectual disability, dysmorphic facial features, and postaxial polydactyly.

Variants of the *PTCH1* gene were first reported in 1996 [9]. Currently, there are 569 *PTCH1* variants reported in the Leiden Open Variation Database (LOVD, version 3.0, https://databases.lovd.nl/shared/variants/PTCH1, accessed on 18 May 2023). Of the reported variants, only 69 are pathogenic or likely pathogenic, and the rest are classified as having unknown significance or as being benign. The ClinVar database reports 393 pathogenic or likely pathogenic *PTCH1* variants.

The first *SUFU* variant associated with BCNS was published in 2009. Currently, 28 pathogenic or likely pathogenic unique variants are reported for *SUFU* in LOVD

(https://databases.lovd.nl/shared/variants/SUFU/unique, accessed on 18 May 2023) and 58 unique variants in the ClinVar database.

The first suggestion of a pathogenic role for *PTCH2* in BCNS was published in 1999 [10]. Since the *PTCH2* gene encodes a protein that is highly homologous to PTCH1, the question emerged as to whether it is also a disease-causing gene in BCNS. Using single-stranded conformational polymorphism analysis, a truncating mutation of the *PTCH2* gene was identified in a patient with medulloblastoma. Subsequently, a few additional publications described cases in which variants of *PTCH2* were linked to BCNS. However, direct evidence to support this proposed association is lacking. To date, no pathogenic *PTCH2* variants associated with the BCNS phenotype have been reported in either the LOVD (https://databases.lovd.nl/shared/variants/PTCH2/unique, accessed on 18 May 2023) or the ClinVar database. Thus, it is not known whether *PTCH2*-caused BCNS is, for example, a rarely reported subtype with a milder phenotype than BCNS arising from other mutations [11] or *PTCH2* does not predispose to BCNS but, instead, is a phenotype modifier [12]. Somatic mutations in *PTCH2* have been implicated in the development of BCC1 (OMIM 605462) and medulloblastoma (OMIM 155255).

Here, we aimed to elucidate the genetic background of BCNS in a Hungarian cohort that included 11 affected families. Whole-exome sequencing (WES) and multiplex-ligation-dependent probe amplification (MLPA) were used to screen pathogenic variants in the known disease-causing genes (*PTCH1* and *SUFU*) and *PTCH2*. With our results, we hoped to widen the pathogenic variant spectrum of genes causing BCNS and also contribute to the establishment of genotype–phenotype correlations. We also aimed to determine how effective our genetic-screening design is for identifying mutations in the genes included for this cohort and, thus, the percentage of cases for which mutations in these genes cause the disease. Our results confirm that missing heritability, which is often an issue for complex diseases, could be a significant phenomenon in rare monogenic diseases, such as BCNS.

## 2. Results

In our Hungarian BCNS cohort (Table 1), WES identified two novel likely pathogenic mutations of the *PTCH1* (NM_000264.5) gene: the c.2994C>A p.Cys998Ter heterozygous nonsense mutation in patient 12 and the c.814_818del p.Asn272SerfsTer11 heterozygous frameshift mutation in patients numbers 13 and 14 (Figure 1, Table 2). The heterozygous frameshift variant (p.Asn272SerfsTer11) is located in the sixth exon of the NM_000264.5 variant, which has 24 exons. Based on the American College of Medical Genetics (ACMG) variant classification guideline [13], this variant is classified as pathogenic, as null variants (frameshift) in the *PTCH1* gene are predicted to cause loss-of-function (a known mechanism of the disease). The affected exon contains 10 additional pathogenic frameshift variants (PVS1) in database. The identified frameshift variant is not present in the gnomAD population database (accessed on 18 May 2023) (PM2). The identified frameshift variant results in a premature termination codon occurring in 11 amino acids downstream of the frameshift, which may either cause nonsense-mediated RNA decay or the production of a severely mutated protein that is missing in most of the 3′ ends. Therefore, we hypothesized that the identified novel variant has a severe loss-of-function impact on protein functions. The other novel variant (p.Cys998Ter) is a heterozygous nonsense variant in the eighteenth exon of the *PTCH1* gene. Based on the ACMG variant classification guidelines [13], this variant is also classified as pathogenic, as null variants (nonsense) in the *PTCH1* gene are predicted to cause loss-of-function (a known mechanism of the disease). The affected exon contains eight other pathogenic nonsense variants (PVS1). The identified nonsense variant, which is not present in the gnomAD population database (PM2), results in the formation of a premature termination codon, likely causing nonsense-mediated RNA decay or a severely mutated protein that is also missing much of the 3′ ends of the protein. Therefore, we hypothesized that the identified novel variant has a severe loss-of-function impact on protein functions. Based on the ACMG guidelines and the results from the analysis with the Franklin Variant Effect Predictor (www.franklin.genoox.com, accessed on 18 May 2023), these novel variants are considered to be disease-causing in relation to BCNS.

**Table 1.** Summary of the clinical phenotypes presented by the investigated BCNS patients.

| Patient Number | Major Criteria | | | | | | Minor Criteria | | | Result of the Genetic Screen |
|---|---|---|---|---|---|---|---|---|---|---|
| | Odontogenic Keratocysts of the Jaw | Multiple BCCs | Calcification of the Falx Cerebri | Pits | Bifid, Fused Ribs | First-Degree Relative with BCNS (Patient Number) | Macrocephaly | Congenital Malformations | Polydactyly | |
| 1 | + | | + | | | | | + | | No variant identified |
| 2 | + | | + | | | | | | | No variant identified |
| 3 | | + | | + | | 4.5 | | | | No variant identified |
| 4 | + | + | + | + | | 3.5 | | | | No variant identified |
| 5 | + | + | | + | | 3.4 | | | | No variant identified |
| 6 | + | | | | | 7 | | + | | No variant identified |
| 7 | + | + | + | | | 6 | + | + | | No variant identified |
| 8 | + | + | | | | | | | | p.Leu297Pro |
| 9 | + | + | | | | | | | | p.Gln527Ter |
| 10 | + | + | | | | 11 | | | | p.Q714Ter |
| 11 | | | | | | 10 | | + | | p.Q714Ter |
| 12 | + | | | | | | + | + | + | p.Cys998Ter |
| 13 | + | + | | | | 14 | | | | p.Asn272SerfsTer11 |
| 14 | + | | | | | 13 | | | | p.Asn272SerfsTer11 |
| 15 | + | + | | | | | + | + | | DelEx2 |
| 16 | + | | | | + | | + | + | | p.Val580_Val582del |

+ indicates the presence of the trait.

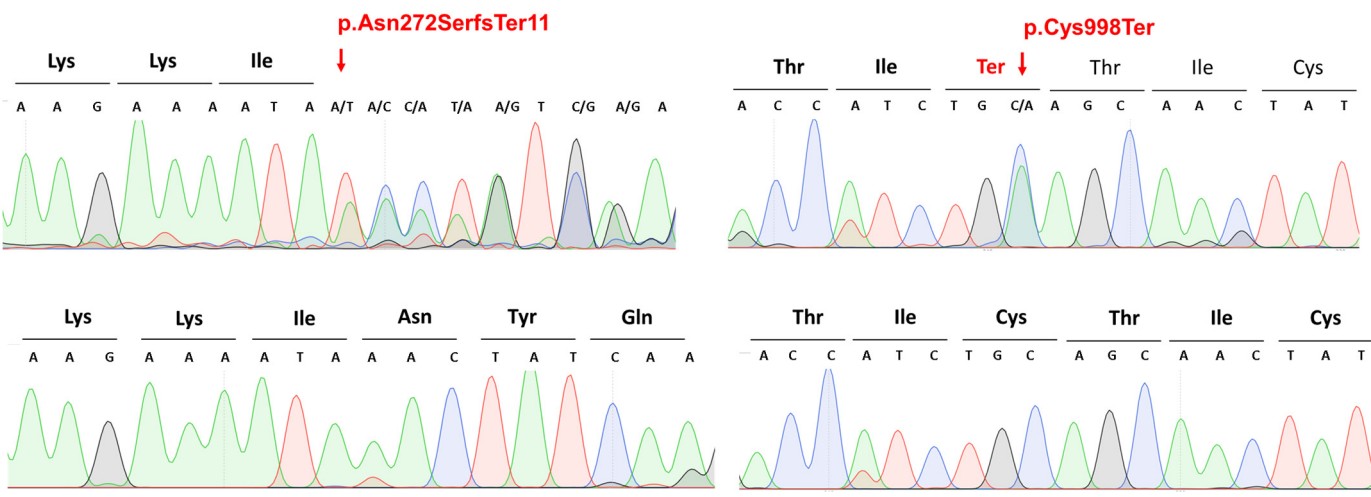

**Figure 1.** Sequencing results of the two novel mutations of the *PTCH1* gene.

**Table 2.** ACMG classification of the two novel variants.

| | ACMG | ACMG Classification |
|---|---|---|
| **PTCH1:c.814_818delAACTA** | Likely Pathogenic | **PVS1, PM2** |
| **PTCH1(NM_000264.5):c.2994C>A** | Likely Pathogenic | **PVS1, PM2** |
| **Functional Predictions** | **PTCH1:c.2994C>A** | **PTCH1:c.814_818delAACTA** |
| **Functional Coding** | | |
| MT | Deleterious (1) | (N/A) |
| DANN | Deleterious (0.99) | (N/A) |
| BayesDel | Deleterious (Strong) (0.6) | (N/A) |
| **Splice Altering** | | |
| SpliceAI | Uncertain (0.11) | (N/A) |
| **Conservation** | | (N/A) |
| GERP | Uncertain (3.05) | (N/A) |
| Functional Whole Genome | | (N/A) |
| GenoCanyon | Deleterious (1) | (N/A) |
| fitCons | Deleterious (0.71) | (N/A) |
| CADD | 36 | |

ACMG: variant classification according to the guidelines of the American College of Medical Genetics and Genomics [13]; N/A indicates not available.

WES also revealed four recurrent pathogenic variants of the *PTCH1* gene: the c.1737_1745del p.Val580_Val582del heterozygous deletion in patient 16 (previously published by our research group [14]), the c.890T>C p.Leu297Pro heterozygous missense mutation in patient 8, the c.1579C>T p.Gln527Ter heterozygous nonsense mutation in patient 9, and the c.2140C>T p.Gln714Ter heterozygous nonsense variant in patients 10 and 11 (Table 3).

All mutations identified via WES were validated with Sanger sequencing.

From the MLPA results, a pathogenic deletion of the second exon of the *PTCH1* gene was identified in patient 15 (Figure 2). WES did not identify pathogenic variants of the *PTCH1* gene for this patient.

**Table 3.** Summary of results from the genetic screening of the investigated BCNS patients.

| Gene | cDNA Variant | Protein Variant | Clinical Significance | Novelty |
|---|---|---|---|---|
| *PTCH1* | c.814_818del | p.Asn272SerfsTer11 | Likely pathogenic | Newly identified by this study |
| *PTCH1* | c.1737_1745del | p.Val580_Val582del | Likely pathogenic | Recently identified by our research group |
| *PTCH1* | c.2994C>A | p.Cys998Ter | Likely pathogenic | Newly identified by this study |
| *PTCH1* | c.890T>C | p.Leu297Pro | Likely pathogenic | Recurrent |
| *PTCH1* | c.1579C>T | p.Gln527Ter | Pathogenic | Recurrent |
| *PTCH1* | c.2140C>T | p.Q714Ter | Pathogenic | Recurrent |

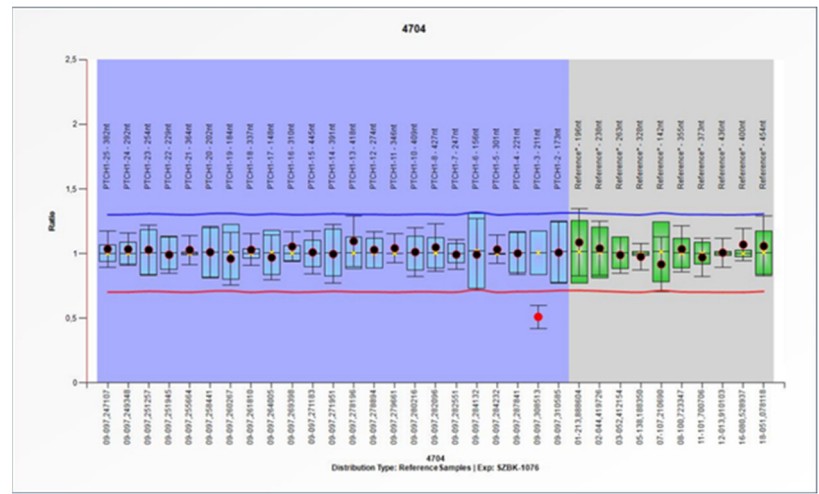

**Figure 2.** MLPA analysis of the *PTCH1* gene in patient 15.

Causative mutations in the *SUFU* and *PTCH2* genes were not detected in our cohort.

The genetic backgrounds of seven of the eleven investigated BCNS families were determined (63.6% diagnostic yield), and correspondingly, missing heritability accounted for 36.36% in this Hungarian cohort. We found a significant association between the presence of *PTCH1* pathogenic variants and the presence of the BCNS clinical phenotype (Chi2 test: $p = 0.0041$). Statistical analyses were carried out using VassarStats (http://faculty. vassar.edu/lowry/VassarStats.html, accessed on 18 May 2023).

## 3. Discussion

Here, we report two novel and four recurrent variants of the *PTCH1* gene in a Hungarian BCNS patient cohort using the WES and MLPA methods. The *PTCH1* gene encodes a 1447 amino acid transmembrane protein (Q13635), which has 12 known transmembrane regions and six extracellular, five intracellular, N-terminal and C-terminal domains (Figure 3) [15]. The identified c.814_818del p.Asn272SerfsTer11 novel variant is located in the first large extracellular loop, whereas the c.2994C>A p.Cys998Ter novel variant is located in the second large extracellular loop of the PTCH1 protein (Figure 3). Both of these large extracellular loops are required for the binding of N-SHH to the patched protein [16]. These mutations are located in evolutionarily conserved regions of the PTCH1 protein (AMIN-ODE evolutionary analysis, www.aminode.org, accessed on 18 May 2023) (Figure 3c). Both truncating variants are predicted in silico to cause early nonsense-mediated mRNA decay and, thus, most probably lead to haploinsufficiency or loss of function (NMDEscPredictor, shinyapps.io, accessed on 18 May 2023).

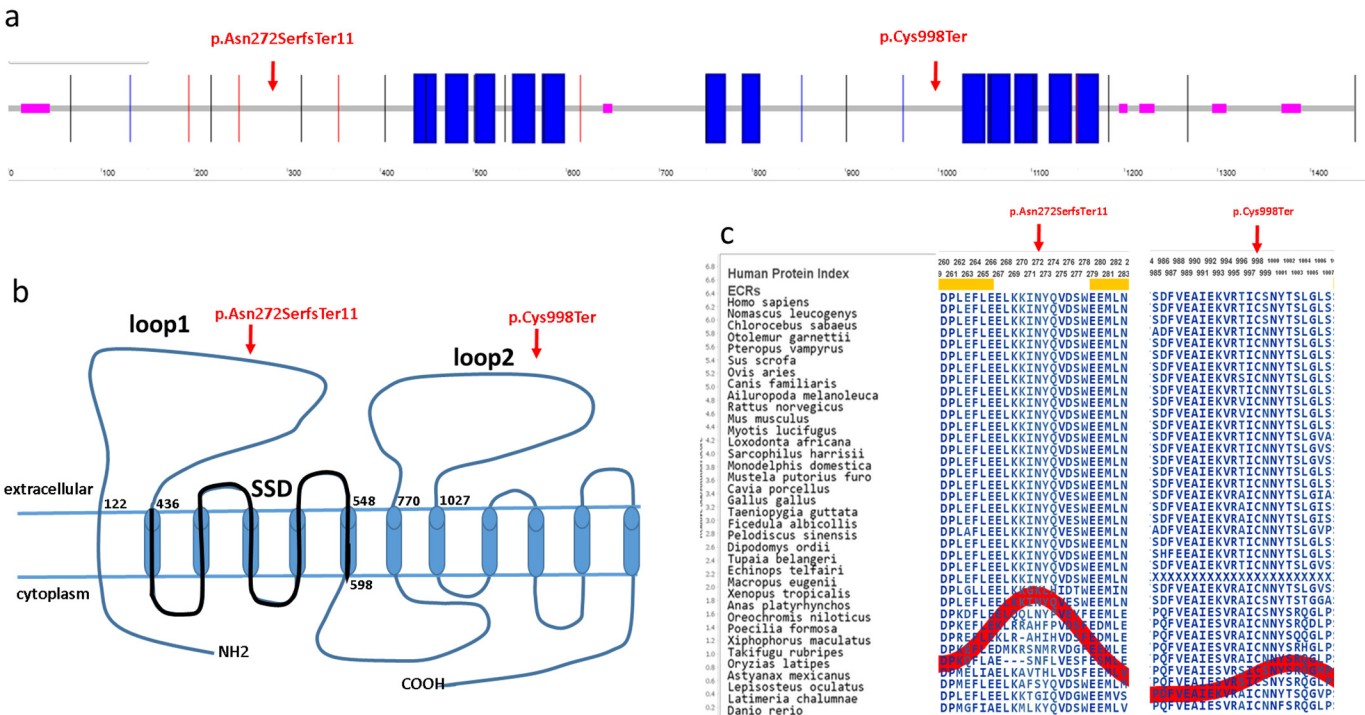

**Figure 3.** Location of the novel variants in the PTCH1 protein. (**a**) Schematic representation of the PTCH1 protein (SMART protein, http://smart.embl-heidelberg.de/, accessed on 18 May 2023). (**b**) Predicted structure of the PTCH1 protein [17]. Thick line denotes the sterol sensing domain (SDD). (**c**) Evolutionary conservation of the affected p.Asn272SerfsTer11 and p.Cys998Ter regions (Aminode, http://www.aminode.org/search, accessed on 18 May 2023). The red line represents the relative rate of amino acid substitution calculated at each protein position.

Pathogenic mutations in the *PTCH1* gene have been found in all domains, but mutations occur most frequently in the first and second large extracellular loops and in the third intracellular domain [15]. The identified recurrent p.Leu297Pro variant is a likely pathogenic missense variant, which was previously reported in the Netherlands [18]. The recurrent p.Gln714Ter pathogenic nonsense variant was identified in Australia [19], and the p.Gln527Ter pathogenic nonsense variant occurs relatively frequently and is reported worldwide [20].

The MLPA technique was used to investigate the possibility of large deletions in the *PTCH1* gene. The method identified a deletion of exon 2. As WES did not identify any pathogenic variants of the *PTCH1* gene in the MLPA-positive patient, our result suggests the importance of this combination of screening methods for the genetic diagnosis of BCNS.

*PTCH1* mutations reported for BCNS patients are predominantly nonsense or frameshift mutations (64%), followed in frequency by splice site mutations (13%), large insertions or deletions (12%), and missense mutations (8%) [21]. In good agreement with these observations, our study identified four nonsense variants, one large deletion, one small deletion and one missense mutation.

Tumors in patients with BCNS are believed to develop according to the two-hit hypothesis, also known as the Knudson hypothesis [22]. The patients usually carry one heterozygous germline variant of the PTCH1 or SUFU genes. To develop the tumors, most tumor suppressor genes require both alleles to be inactivated. The inactivation of the remaining normal allele is usually the consequence of "second hit" somatic variants of the *PTCH1* or *SUFU* gene. Second-hit mutations can be induced by, for example, ultraviolet light or ionizing radiation. In addition to the standard two-hit model, haploinsufficiency, dominant-negative isoforms and epigenetic silencing may be involved in the development of the tumors [23]. Patients with *SUFU* pathogenic variants are reported mostly with medul-

loblastoma and without maxillary keratocysts [4]. Our results are in good agreement with this observation since none of the investigated patients presented with medulloblastoma and none had causative variants in *SUFU*.

The pathogenetic role of *PTCH2* variants in BCNS has not yet been proven. As our study did not identify any pathogenic variants of the *PTCH2* gene, our study also does not support a direct involvement of *PTCH2* variation in the development of BCNS [12].

WES analysis is usually not able to detect non-coding variants further than $+/- 20$ base pairs from exon–intron boundaries. However, intronic or even deep intronic mutations can create strong cryptic splice acceptor sites leading to frameshift mutations, which can lead to premature stop codons. The identification of deep intronic splicing mutations is challenging. Bholah and collages demonstrated the importance of comprehensive transcript mutation analysis for individuals with clinically diagnosed Gorlin syndrome when *PTCH1* variants have not been detected by genomic sequencing or by copy number analysis [24]. Pathogenic mutations can occur deep within the introns of disease-causing genes. Deleterious DNA variants located more than 100 base pairs away from exon–intron junctions most commonly lead to pseudo-exon inclusion due to the activation of non-canonical splice sites or changes in splicing regulatory elements. Additionally, deep intronic mutations can disrupt transcription regulatory motifs and non-coding RNA genes [25].

Since 1996, when the first pathogenic mutations were identified in *PTCH1* [9], an extremely high number of disease-causing variants have been reported for this gene. These discoveries reflect the enormous development in the field of mutation screening technologies. In this study, WES and MLPA were successfully applied to identify the genetic background of approximately 56% of the investigated BCNS patients. A recent study (2022) with a similar approach from a group in the UK determined the genetic background of 75% of the patients in their study [26]. These results further emphasize that even high-throughput genetic screening methods, such as WES, have limitations and that missing heritability is an issue for rare diseases such as BCNS [26].

Missing heritability affects both common and rare diseases and is mainly associated with common and complex diseases where promising modern technological advances, such as genome-wide association studies, were unable to uncover the complete genetic mechanism of the disease [26]. Missing heritability in rare diseases can be the consequence of high phenotypic diversity. Such diversity can be caused by strikingly different phenotypes associated with different variants of the same gene as well as high genetic heterogeneity caused by variations in distinct genes that produce similar phenotypes [26]. Missing heritability shows high variability in rare diseases: it can be extremely low, as in skin tumor CYLD cutaneous syndrome, but it can be relatively high, as in BCNS. To resolve missing heritability in rare diseases, novel technological approaches, such as the promising techniques of whole-genome sequencing (WGS) or epigenetic analysis, should be implemented in clinical practice.

Many large-scale genome sequencing projects are ongoing globally, but clinical implementation of the results of these projects is, for the most part, lagging behind [27]. WGS analysis can be used to detect and interpret single-nucleotide variants, insertions and/or deletions, uniparental disomy, copy-number variations, balanced structural variants, and short tandem repeat expansions [27]. The availability of WES and WGS has drastically impacted genetic diagnostics, and the clinical genetics specialty is undergoing rapid development [27]. The clinical application of WGS can contribute to the genetic diagnosis of rare diseases and additionally help identify novel disease-causing genes [27]. In particular, the integration of WGS into the healthcare setting could potentially reduce missing heritability in rare diseases and increase the genetic diagnostic rate of monogenic diseases, including BCNS.

WES has recently been used more and more frequently in diagnosis; however, a significant proportion of patients remain undiagnosed after sequencing their genome. New approaches, based on functional aspects of the genome, including epigenomics, are beginning to emerge [27]. Increasing numbers of reports describe functionally relevant al-

terations of the genome that do not involve mutation of the nucleotide sequence. Moreover, a considerable number of studies reveal the appearance of aberrant epigenetic modifications of nucleic acids in association with the occurrence of diseases, including cancer, diabetes, Alzheimer's disease, and many others. A better understanding of the exact roles of epigenetic modification in biological processes and in human diseases might make it possible to identify biomarkers for diagnosis and treatment. Notably, remarkable efforts have been made to establish technologies to facilitate the accurate detection and mapping of epigenetic modification. In the near future, the integration of epigenetic analysis in healthcare settings might also help to reduce missing heritability for rare diseases and increase the genetic diagnostic rate in diseases such as BCNS.

Based on our results, we hypothesize that the application of WGS and/or an epigenetic approach should be applied to unsolved BCNS cases to attempt to resolve missing heritability for BCNS.

## 4. Materials and Methods

### 4.1. Patients

Sixteen Hungarian patients from 11 families fulfilling the diagnostic criteria of BCNS were enrolled in this study. Among the patients, nine were female and seven were male. The clinical phenotypes of the affected patients are summarized in Table 1. The most frequent clinical manifestations were histologically proven odontogenic keratocyst of the jaw (13 patients, 81%), multiple BBSC (nine patients, 56%) and congenital malformations in (seven patients, 43%). Bilamellar calcification of the falx cerebri and macrocephaly were detected in four patients (25%). Palmar and plantar pits were present in three patients (18%). Bifid, fused ribs were present in one patient as was polydactyly. Fifty percent of the investigated patients were aware of first-degree relatives affected by BCNS.

Written informed consent was obtained from all the enrolled patients according to a protocol approved by the Hungarian National Public Health Center, in adherence to the Helsinki guidelines. All the enrolled individuals underwent pre- and post-test genetic counselling at the Department of Medical Genetics, University of Szeged (Szeged, Hungary).

### 4.2. DNA Extraction

Genomic DNA was extracted from venous blood mixed with the anticoagulant EDTA using the DNeasy® Blood & Tissue Kit (QIAGEN, Germany), as described in the manufacturer's instructions. For quantification Qubit Fluorometric Quantification instrument was used according to the manufacturer's instructions.

### 4.3. Whole-Exome Sequencing

Genotypes of patients were determined using next-generation sequencing. Library preparation was carried out using the SureSelectQXT Reagent Kit (Agilent Technologies, Santa Clara, CA, USA). Pooled libraries were sequenced on an Illumina NextSeq 550 NGS platform using the 300-cycle Mid Output Kit v2.5 (Illumina, Inc., San Diego, CA, USA). Adapter-trimmed and Q30-filtered paired-end reads were aligned to the hg19 Human Reference Genome using the Burrows–Wheeler Aligner (BWA). Duplicates were marked using the Picard software package. The Genome Analysis Toolkit (GATK) was used for variant calling (BaseSpace BWA Enrichment Workflow v2.1.1. with BWA 0.7.7-isis-1.0.0, Picard: 1.79 and GATK v1.6-23-gf0210b3).

The mean on-target coverage achieved from sequencing was $71\times$ per base, with an average percentage of targets covered greater or equal to $30\times$ of 96% and 90%, respectively. Variants passed by the GATK filter were used for downstream analysis and annotated using ANNOVAR software tool (version 2017 July 17) [28]. Single-nucleotide polymorphism testing was performed as follows: high-quality sequences were aligned with the human reference genome (GRCh37/hg19) to detect sequence variants, and the detected variations were analyzed and annotated. Variants were filtered according to read depth, allele frequency and prevalence in genomic variant databases, such asExAc (v.0.3) and

Kaviar. Variant prioritization tools (PolyPhen2, SIFT, LRT, Mutation Taster, and Mutation Assessor) were used to predict the functional impact. For variant filtering and interpretation, VarSome [29] and Franklin bioinformatic platforms [https://franklin.genoox.com, accessed on 18 May 2023] were used according to the guidelines of the ACMG [13].

All the identified candidate variants were confirmed via bidirectional capillary sequencing. PCR amplification was set up using DreamTaq™ Green PCR Master Mix ready-to-use solution (Thermo Scientific™), as described in the manufacturer's instructions. Reaction conditions were as follows: initial denaturation at 95 °C for 1 min, denaturation at 95 °C for 30 seconds, annealing at 58 °C for 30 seconds, extension at 72 °C for 30 seconds with 35 repeated cycles, and a final extension at 72 °C for 10 min. The following primers were used:

Exon6 Fw: 5′ ctacaaggtggatgcagtgg 3′
Exon6 Rev: 5′ aagtgaacgatgaatggacac 3′
Exon11 Fw: 5′ gctggtggcagagtcctaac 3′
Exon11 Rev: 5′ gcagccagtgacacatcatc 3′
Exon14 Fw: 5′ atgggtattctccgtacaca 3′
Exon14 Rev: 5′ gaagcaatctgatgaactccaaa 3′
Exon18 Fw: 5′ aaaggcctggaggctatga 3′
Exon18 Rev: 5′ gcccagacataaacaaaactt 3′

### 4.4. Multiplex-Ligation-Dependent Probe Amplification

To assess larger genetic aberrations, we used SALSA MLPA Probemix P067 PTCH1 (MRC-Holland, Netherlands) containing probes for 23 of the 25 exons in the *PTCH1* gene (LRG_515; no probes are included for exons 1 and 9), according to the manufacturer's instructions. Amplicon fragment length analysis was performed on an ABI 3500 Genetic Analyzer (ThermoFisher Scientific, Waltham, MA, USA) and analyzed using Coffalyser.net software (MRC-Holland, Amsterdam, The Netherlands).

## 5. Conclusions

BCNS is a familial cancer syndrome, and 85% of the cases develop as a consequence of mutations of the *PTCH1* gene. Using WES, two novel pathogenic mutations were identified in Hungarian BCNS patients. However, applying WES and MLPA together could not identify the genetic background of BCNS in all of the investigated cases, suggesting that even high-throughput genetic screening methods have limitations in the full discovery of the genetic background of this inherited disease. To resolve missing heritability in rare diseases such as BCNS, the application of additional high-throughput genetic methods or non-genetic approaches should be considered in future studies.

**Author Contributions:** Conceptualization, N.N., J.O., M.S. and J.P.; methodology, N.N. and D.N.; validation, N.N.; B.A.B. and D.N.; investigation, M.P. and É.V.; resources, É.V., L.S., E.H. and A.V.; data curation, N.N.; writing, original draft preparation, N.N., M.P. and É.V.; writing, review and editing, D.N. and M.S.; visualization, N.N., M.P. and É.V.; supervision, J.O., M.S. and J.P.; funding acquisition, M.S. All authors have read and agreed to the published version of the manuscript.

**Funding:** This research was supported by the EFOP-3.6.1-16-2016-00008 grant and by the GINOP-2.3.2-15-2016-00039 grant.

**Institutional Review Board Statement:** The study was conducted according to the guidelines of the Declaration of Helsinki and was approved by the Ethics Committee of the University of Szeged (58523-4/2017/EKU; 19.02.2018.;).

**Informed Consent Statement:** Written informed consent was obtained from all subjects involved in the study.

**Data Availability Statement:** Data is available from the authors upon request.

**Acknowledgments:** We thank Dalma Füstös, Zsuzsanna Horváth-Gárgyán, and Anikó Gárgyán for their skilled technical assistance and Shannon Frances for providing language help.

**Conflicts of Interest:** The authors declare no conflict of interest.

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
