# Peer review of "Whole-Exome Sequencing Identified Two Novel Pathogenic Mutations in the PTCH1 Gene in BCNS"

_cimb, doi:10.3390/cimb45070336_

Round 1
Reviewer 1 Report
In this article Margit Paul et al. studied the genetic background of BCNS in a Hungarian cohort and used WES and MLPA to screen for PTCH1, PTCH2 and SUFU genes.
The paragraphs are well written. The topic is good and the article could be improved by better flow, and some editing.
1. Usually, the figures need a short explanation. Figure 2 has no caption
2. The selection criteria of 16 patients in the study is not clear. It is usually based on statistical calculations, but no standard is specified
3. Since this study was conducted in Hungary, it is better to compare it with similar studies in other countries or ethnic groups in the discussion section.
Reviewer 2 Report
The authors identified two novel likely pathogenic mutations of the PTCH1 in Hungarian BCNS patients, which will lead to the formation of premature stop codon.
No major issues have been found. But there are some minor issues that may need to be addressed.
1> It will be better if you can validate the novel mutations from your analysis via experiment if possible
2> In the methodology part, you should specify the important parameters set up for WES analysis.
3> It will be better if you can use statistical method to explore the correlations between results of genetic screen and criteria of BCNS.
